# Chronic Fatigue Exhibits Heterogeneous Autoimmunity Characteristics Which Reflect Etiology

**Olga V. Danilenko [1], Natalia Y. Gavrilova [2,3,4,\*] and Leonid P. Churilov [1,5]**

1   Department of Pathology, Faculty of Medicine, Saint Petersburg State University, Universitetskaya emb, 7–9, 199034 St. Petersburg, Russia; olgadanil@mail.ru (O.V.D.); elpach@mail.ru (L.P.C.)
2   Department of Faculty Therapy, Faculty of Medicine, Saint Petersburg State University, Universitetskaya emb, 7–9, 199034 St. Petersburg, Russia
3   Department of Pulmonology, Saint Petersburg Research Institute of Phthisiopulmonology, Ligovskii pr., 2–4, 191036 St. Petersburg, Russia
4   Research Institute of Phthisiopulmonology, St. Petersburg State University Hospital, 190103, Fontanka River Emb., 154, St. Petersburg, Russia
5   Department of Experimental Tuberculosis and Innovative Technologies, Saint Petersburg Research Institute of Phthisiopulmonology, Ligovskii pr., 2-4, 191036 St. Petersburg, Russia
\*   Correspondence: fromrussiawithlove_nb@mail.ru

**Abstract:** Chronic fatigue syndrome/myalgic encephalomyelitis (CFS/ME) is considered to be associated with post-viral complications and mental stress, but the role of autoimmunity also remains promising. A comparison of autoimmune profiles in chronic fatigue of different origin may bring insights on the pathogenesis of this disease. Thirty-three patients with CFS/ME were divided into three subgroups. The first group included Herpesviridae carriers (group V), the second group included stress-related causes of chronic fatigue (distress, group D), and the third group included idiopathic CFS/ME (group I). Were evaluated thirty-six neural and visceral autoantigens with the ELISA ELI-test (Biomarker, Russia) and compared to 20 healthy donors, either without any fatigue (group H), or "healthy but tired" (group HTd) with episodes of fatigue related to job burnout not fitting the CFS/ME criteria. β2-glycoprotein-I autoantibodies were increased in CFS/ME patients, but not in healthy participants, that alludes the link between CFS/ME and antiphospholipid syndrome (APS) earlier suspected by Berg et al. (1999). In CFS/ME patients, an increase in levels of autoantibodies towards the non-specific components of tissue debris (double-stranded DNA, collagen) was shown. Both CFS and HTd subgroups had elevated level of autoantibodies against serotonin receptors, glial fibrillary acidic protein and protein S100. Only group V showed an elevation in the autoantibodies towards voltage-gated calcium channels, and only group D had elevated levels of dopamine-, glutamate- and GABA-receptor autoantibodies, as well as NF200-protein autoantibodies. Therefore, increased autoimmune reactions to the multiple neural antigens and to adrenal medullar antigen, but not to other tissue-specific somatic ones were revealed. An increase in autoantibody levels towards some neural and non-tissue-specific antigens strongly correlated with a CFS/ME diagnosis. Autoimmune reactions were described in all subtypes of the clinically significant chronic fatigue. Visceral complaints in CFS/ME patients may be secondary to the neuroendocrine involvement and autoimmune dysautonomia. CFS may be closely interrelated with antiphospholipid syndrome, that requires further study.

**Keywords:** autoimmunity; anti-receptor autoantibodies; antiphospholipid syndrome; chronic fatigue syndrome/myalgic encephalomyelitis; dysautonomia; Herpesviridae; stress

## 1. Introduction

Chronic fatigue syndrome/myalgic encephalomyelitis (CFS/ME), mentioned in the ICD-10 classification as "R53.82" (unspecified fatigue) or as "G93.3" (post-viral fatigue), is a heterogeneous entity that manifests with the pronounced disabling fatigue without relief after rest, accompanied by sleep disturbances, and cognitive impairment [1,2]. The most common symptoms are the aggravation of the fatigue to the point of exhaustion after physical or mental effort (post-exertional malaise), diffuse pain, neuroendocrine disorders, immune dysfunction and dysautonomia [3,4]. However, the core symptom of CFS—long-lasting fatigue that cannot be relieved enough by sleep and rest—also is a non-specific widespread manifestation in a long list of other somatic and neurological diseases. The incidence of CFS, after excluding other causes of clinically significant chronic fatigue, remains very high and, according to the recent studies, affects about 0.89% of the world population [5]. Some studies allude that chronic fatigue may result from a neuro-immuno-endocrine dysfunction [6].

Substantial evidence for the role of autoimmunity in CFS has been reported recently. Genetic similarities between CFS and autoimmune disorders, association of CFS onset with the autoimmune-associated exogenous risk factors, dysfunction of several immune cell subsets in CFS, its comorbidity with other autoimmune diseases, and, finally, the hyperproduction of various autoantibodies (AAb) in CFS patients contribute to this hypothesis [7–10].

Among the different types of AAb, a group of AAb against G-protein-coupled receptors has been evaluated in CFS. Higher levels of AAb against M1, M3, and M4 acetylcholine receptor (AChR) and $\beta2$ adrenergic receptor (AdR) were described in CFS patients compared to healthy controls [11]. Their pathophysiological relevance is supported by the clinical evidence, including the removal of anti-$\beta2$ AdR and anti-M3/M4 AChR AAb in CFS by the immunoabsorption method, followed by rapid clinical improvement [12]. The evidence for the dysfunction of the hypothalamic–pituitary–adrenal axis (HPA) in CFS may also support this theory, because the hypothalamus is both a supreme unit of the autonomic nervous system and neuroendocrine interface [13]. Immuno-inflammatory pathways were shown to potentially down-regulate the function of the HPA axis in CFS [14]. The activation of microglia in CFS was observed by positron emission tomography, and the signals in the amygdala, thalamus, and midbrain positively correlated with the cognitive impairment score, those in the cingulate cortex and thalamus, with pain score, and in the hippocampus, with depression score [15]. To our best knowledge, CFS was first hypothesized to be an "autoimmune chronic hypothalamitis" by A.Sh. Zaichik and L.P. Churilov in 1999 [16]. Later, this hypothesis of CFS as an autoimmune/inflammatory disorder of the hypothalamus was shared by other authors [14,17]. An experimental model of CFS, created by animal immunization with synthetic analogues of viral polyribonucleotides, demonstrated signs of neuroinflammation, glial activation, and serotonin reuptake transporter failure. The impairments to the hypothalamic–pituitary–adrenal system were reported in this model as well [18]. In particular, a decrease in the adrenocorticotropic hormone sensitivity of adrenal cells and suppression of the negative feedback mechanism were detected [19]. Although some researchers did not find any differences between the levels of antineuronal AAb in CFS and healthy individuals [20], others found a decrease of AAb towards glial fibrillary acidic protein in remission, and an increase in exacerbations of the disease, correlated with the presence of Epstein-Barr virus [21].

A role of antiphospholipid antibodies and the link between the APS and the CFS/ME development remains to be an important question. Hokama et al. in their study described 41 patients with CFS, Gulf War syndrome and chronic Ciguatera fish poisoning and evauated 37 sera (90.2%) positive for anticardiolipin AAb [22]. In their other study, immunoglobulin M isotypes of antiphospholipid AAb was evaluated in 95% of CFS patients [23]. Berg et al. showed decreased coagulation activation from immunoglobulins (Igs) and high titers of anti-B2GPI AAb, that allude authors to classify CFS/ME and fibromyalgia as a type of antiphospholipid antibody syndrome.

While both mast cells [17] and the innate immune system [24] were regarded as triggers for the focal inflammation in the hypothalamus in CFS, the role of the adaptive immune system should also be considered. The concept of "autoimmune hypothalamopathy", which results from the functional effects of anti-G-protein-coupled receptors AAb on the function of both AdR and AChR, appears to be promising in CFS. The ability of serum AAb against the muscarinic AChR to affect the brain cholinergic system has been proven with positron emission tomography [25]. Nevertheless, the spectrum and intensity of autoimmunity in CFS is not entirely elucidated until now.

Various AAb in low titers may be described in healthy individuals as well. Hence, just the presence of certain AAb in CFS or other diseases cannot be interpreted as a sign of disease [26]. The whole spectrum of immune autoreactivity should be described in CFS patients and compared to healthy individuals.

Clinically significant chronic fatigue without manifestations of primary therapeutic or neuroendocrine diseases can be associated with various etiological factors, such as chronic stress (F48.0, neurasthenia due to distress), or viral infection complications, in particular caused by Herpesviridae (G93.3, post-viral asthenia). Chronic fatigue may also be "idiopathic" (R53.83—chronic malaise and fatigue, not otherwise classified). In ICD 11, the transition to which is recommended by the World Health Organization from 1 January 2022, a separate code "postviral fatigue syndrome" is preserved (8E49), but it is located under the heading "Other disorders of the nervous system", and not only "myalgic encephalomyelitis" but also "chronic fatigue syndrome" are now listed as included diagnostic terms for this code [27].

Despite the presence of several sets of validated criteria for the diagnosis of CFS, widely used in the world [28–30], low awareness of the medical specialists and lack of reliable laboratory diagnostic markers are possibly the main reasons that up to 80% of cases of CFS remain unrecognized [31]. Therefore, the comparison of the autoimmunity spectra in all of the above-mentioned cases of chronic fatigue may be of considerable importance both for the diagnosis and the treatment purposes.

## 2. Material and Methods

The study involved 53 individuals, including 33 patients with clinically significant chronic fatigue who met the CFS/ME criteria. The informed consent was signed by all participants. Diagnosis of CFS was verified using clinical and laboratory criteria of the Centers for Disease Control (USA, 1994), in a simplified version from 2005 [30–32]. In all patients the presence and severity of anxiety and depression associated with fatigue was assessed using the Hospital Anxiety and Depression Scale (HADS) [33]. Chronic fatigue was considered as having a clinically significant impact on a patient's life with a HADS score of ≥11. All subjects did not have any established rheumatological or endocrinological diagnosis and did not mention in anamnesis any specific daily regimen, lifestyle or diet modifications that could mimic the CFS symptoms. At the time of examination, they were not suffering from any acute disease or exacerbation of chronic ones, e.g., patients were not suffering from acute viral respiratory diseases or intestinal infections, not only at the time of examination, but also for at least six weeks prior to it. The study did not include patients who had been vaccinated less than 3 months before the survey, as well as those who had addictive habits at the time of the survey (or had abandoned them less than a year ago). Individuals who worked in hazardous industries for any period of time less than 5 years before the examination were excluded, as well as pregnant and breast-feeding women. All patients did not use any medications for 4 weeks before blood sampling, except for contraceptives, anti-hypertensive and lipid-lowering drugs.

In all patients the possible presence of a previous infection caused by Herpesviridae was verified both clinically and immunologically (for Epstein-Barr virus, cytomegalovirus, herpes simplex virus types 1 and 2, and human herpesvirus type 6)—by chemiluminescence enzyme immunoassay analysis of blood sera for the presence of IgG and IgM against the virus core antigen. The following reagent kits were used: Vecto HSV-1,2-IgG,

Vecto HSV-IgM, Vecto HHV-6-IgG, Vecto EBV-VCA-IgM, Vecto EBV-NA-IgG, Vecto CMV-IgG, and Vecto CMV-IgM (Vector-Best, Russia). At the same time, all patients included in the study had negative tests for antibodies to the capsid antigens of the above-mentioned viruses and negative polymerase chain reaction analysis for the DNA of these viruses. This part of the study was performed in collaboration with St. Petersburg State Budgetary Healthcare Institution "S.P. Botkin Clinical Infectious Diseases Hospital" affiliated with the Saint Petersburg University as its clinical base.

The presence of the confirmed infection with the various Herpesviridae species was considered reliable in *C exam.* > *C crit.*, where *C exam.* was a concentration of the antiviral antibodies in the serum of the examined patient, and *C crit.* included borderline concentrations of the antiviral antibodies in the control serum, multiplied by the coefficient R, recommended by the kit manufacturer for every particular diagnostic reagent kit.

Those patients, who meet the CFS criteria with a *C exam.* > *C crit.* were included in the G93.3 study group (post-viral asthenia, group V). In cases when *C exam.* = *C crit.*, the results were considered doubtful, and such patients were excluded from the study. If *C exam.* < *C crit.*, the role of Herpesviridae spp. as a causative factor of CFS/ME was considered negative, and these individuals, depending on their anamnesis and prevailing symptoms were included either into the F48.0 group (post-distress neurasthenia, group D) or into R53.83 group (idiopathic chronic fatigue, group I).

The study included 20 participants who remained negative according to the CFS/ME criteria. They were divided into the comparison group and the control group. The control group consisted of healthy participants (Z00.0; clinically healthy individuals, without any complaints and with negative viral tests, n = 12, age 20–30 years, group H). Due to the fact that frequent episodic fatigue, not matching the CFS/ME criteria, is a widespread symptom, especially among elderly people prone to "job burnout", a clinical state that is sometimes compared to CFS [32], we selected a special comparison group (Z73. 0) composed of eight practically healthy individuals, aged 45–55 years old, with negative viral tests, but having complaints of job burnout and recurrent fatigue episodes that did not reach a clinically significant level, with a total score of HADS ≤ 10 ("healthy, but tired", group HTd).

The evaluation of AAb against 36 different neural and visceral autoantigens, as well as non-organ-specific autoantigens was performed using peripheral venous blood sera of patients by non-competitive solid-phase immunoassay (ELI-Test) [33,34]. The following ELI-test kits were applied: "ELI-Viscero-24", "ELI-Neuro-12", and "ELI-Pulmo-6" (Biomarker, Moscow, Russia). The control serum is a preparation of polyclonal immunoglobulins of the IgG class, synthesized by B-lymphocytes in response to antigenic stimuli that occurred throughout the life of donors. Control serum immunoglobulins were obtained from the blood serum of more than 5000 healthy donors and brought to a concentration close to physiological (16 mg/mL).

Thus, this sample contains population-normalized IgG class polyclonal antibodies to each of the studied antigens. This makes it possible to use the control sample as a universal standard for all tested antigens in the test. Depending on the studied antigen, the control sample is diluted to a final concentration, which is calculated (derived) on the basis of studies of the level of autoantibodies of a large cohort of healthy people (individual serum samples). The reaction of the control sample in individual dilutions with different antigens reflects the individual profile of a healthy person in the population in the corresponding age group. When comparing the values of the parameters of the test sample (patient) with the control sample, we obtain a profile of deviations in the content of an individual's autoantibodies from the population norm. The content of AAb to the antigens listed below (Table 1) was evaluated in the conventional units of optical density and relative to their content in a control pool of sera from healthy donors (taken for 100%), as well as relative to the individual average autoimmune reactivity of each individual.

**Table 1.** List of antigens, which autoantibodies were tested.

| № | Antigen | Abbreviation |
|---|---|---|
| 1 | Double stranded deoxyribonucleic acid | ds-DNA |
| 2 | β2-glycoprotein-I | β2GPI |
| 3 | Fc-fragments of IgG | Fc-Ig |
| 4 | Membrane antigen of cardiomyocytes | CoM-0.2 |
| 5 | β1-adrenergic receptors of cardiomyocytes | β1AR |
| 6 | Platelet membrane antigen | TrM-03 |
| 7 | Cytoplasmic antigen of neutrophils | ANCA |
| 8 | Membrane antigen of renal glomerular cells | KiM-05 |
| 9 | Cytoplasmic antigen of renal glomerular cells | KiS-07 |
| 10 | Membrane antigen of pulmonary alveolocytes | LuM-02 |
| 11 | Cytoplasmic antigen of pulmonary alveolocytes with a molecular weight of ~ 80 kDa | LuS-06-80 |
| 12 | Cytoplasmic antigen of pulmonary alveolocytes with a molecular weight of ~ 300 kDa | LuS-300 |
| 13 | Collagen type IV | Collagen |
| 14 | Pulmonary elastin | Elastin |
| 15 | Membrane antigen of gastric wall cells | GaM-02 |
| 16 | Membrane antigen of cells of small intestine wall | ItM-07 |
| 17 | Cytoplasmic antigen of hepatocytes | HeS-08 |
| 18 | Membrane antigen of hepatocyte mitochondria | HMMP |
| 19 | Human insulin | Ins |
| 20 | Insulin receptors | Ins-R |
| 21 | Thyroglobulin | TG |
| 22 | Thyrotropin receptor | TSH-R |
| 23 | Membrane antigen of adrenal medulla cells | AdrM-D/C-0 |
| 24 | Membrane antigen of sperm and prostate cells | Spr-0.6 |
| 25 | γ-interferon | hamma-ifn/hamma-IFN |
| 26 | S100 protein | S100 |
| 27 | Glial fibrillary acidic protein | GFAP |
| 28 | Myelin basic protein | MBP |
| 29 | Voltage-dependent calcium channel | VDCh |
| 30 | N-cholinergic receptors | Hol-R |
| 31 | Serotonin receptors | Ser-R |
| 32 | γ-aminobutyric acid receptors | GABA-R |
| 33 | Dopamine receptors | Da-R |
| 34 | Glutamate receptors | Glu-R |
| 35 | Neurofilament protein 200 | NF-200 |

For each kind of AAb, the following parameters were calculated, according to the manufacturer's recommendations [34–36]:

- Percentage gap compared to the results of the healthy donors' pool sera (with a "+" sign (above the healthy pool value), or with a "-" sign (below the healthy pool value);
- The average autoimmune reactivity of an individual, calculated as the algebraic sum of all deviations from the control healthy donors' pool for each type of AAb, divided by the number of measured autoantibodies;
- The profile of autoimmunity in an individual, representing the variation in the deviations of each AAb level, from the individual average autoimmune reactivity, taken as the isoline.

Statistical processing was performed with the Statistica 10.0 software package using the parametric and nonparametric statistics. Methods of variation statistics based on the

analysis of absolute and relative values were used. Quantitative data was calculated as M ± m, where M is the arithmetic mean, m is the standard error. To compare the profiles of AAb between groups, the Pearson $\chi^2$ test and the Wilcoxon Rank Sum test were performed. The following reference data of descriptive and variation statistics were evaluated: Minimum value (Vmin), maximum value (Vmax), limit (Lim = Vmin–Vmax), confidence interval 95% (CI 95%), confidence interval 75% (CI 75%), mode (Mo), mean value (M), standard deviation (±sd, or M ± sd), excess (E), amplitude of variation spread (Ampl), and coefficient of variation (Cv). Differences were considered significant both according to the Pearson correlation method and using the Fisher coefficient at $p < 0.05$. Spearman correlation analysis, studying the relationship between the presence of chronic fatigue and the parameter "concentration of AAb X" (where X is the concentration of AAb against each of the neural antigens by "ELI-Neuro-12") was also performed. In this case, the measure of clinical risks was determined, that is, the direction and degree of the relationship between the presence of an increased level of certain AAb and positivity for ME/CFS criteria (ANOVA analysis of variance, using the Pearson $\chi^2$ coefficient and the Wilcoxon Rank Sum test).

### 3. Results

In individuals with clinically significant chronic fatigue, their anxiety and depression score based on the HADS scale was significantly elevated (8–14 points, on average: 11 points. The scores were higher ($p < 0.05$) than in healthy donors complaining of the episodes of fatigue that did not meet CFS criteria (2–4 points, on average 3 points (Figure 1).

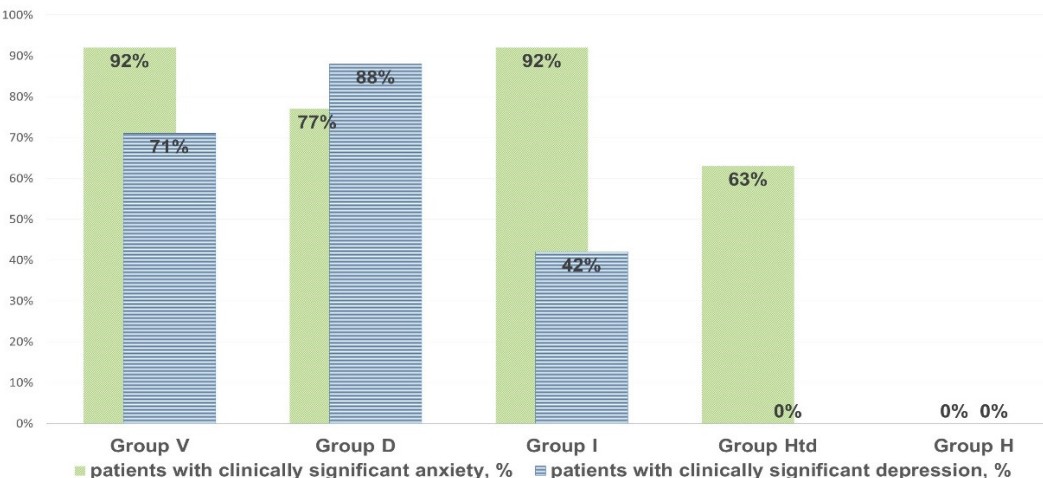

**Figure 1.** The prevalence of clinically significant anxiety and depression (with HADS values > 11) in chronic fatigue patients, in "healthy but tired" and in controls.

The characteristics and intensity of the autoimmune process against neural and non-organ-specific visceral antigens was impaired in all types of chronic fatigue, but there were considerable differences depending on fatigue etiology.

As shown in Figures 2–4, in patients with clinically significant chronic fatigue of post-viral etiology (group V), the relative level of AAb to a number of autoantigens expressed in the nervous tissue (receptors of serotonin, γ-aminobutyric acid and glutamate), as well as Aab towards voltage-dependent calcium channels, was significantly increased compared to that in healthy subjects without complaints of fatigue (group H). In addition, the relative level of AAb to non-organ-specific autoantigens associated with tissue debris and apoptotic bodies (double-stranded DNA, collagen and, especially, β2-glycoprotein-1, a ligand of phospholipids) was also statistically significantly increased in group V ($p < 0.05$).

At the same time, the relative level of autoimmune reactivity against organ-specific visceral antigens in these groups did not differ significantly. In the post-viral chronic fatigue group (group V), relative intensity of autoimmune processes was increased significantly only against the adrenal medullar antigen ($p < 0.05$).

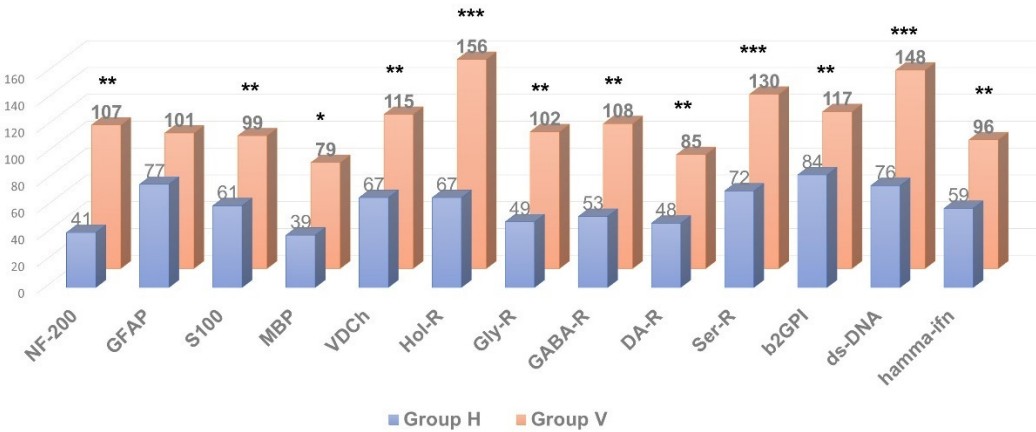

*p≤0.05, **p≤0.01, ***p≤0.001 Group V > Group H

**Figure 2.** Comparative analysis of anti-neural autoantibody concentration in individuals with clinically significant chronic fatigue of post-viral etiology (group V, n = 13) and healthy individuals (group H, n = 12). * $p \le 0.05$, ** $p \le 0.01$, *** $p \le 0.001$, group V > group H.

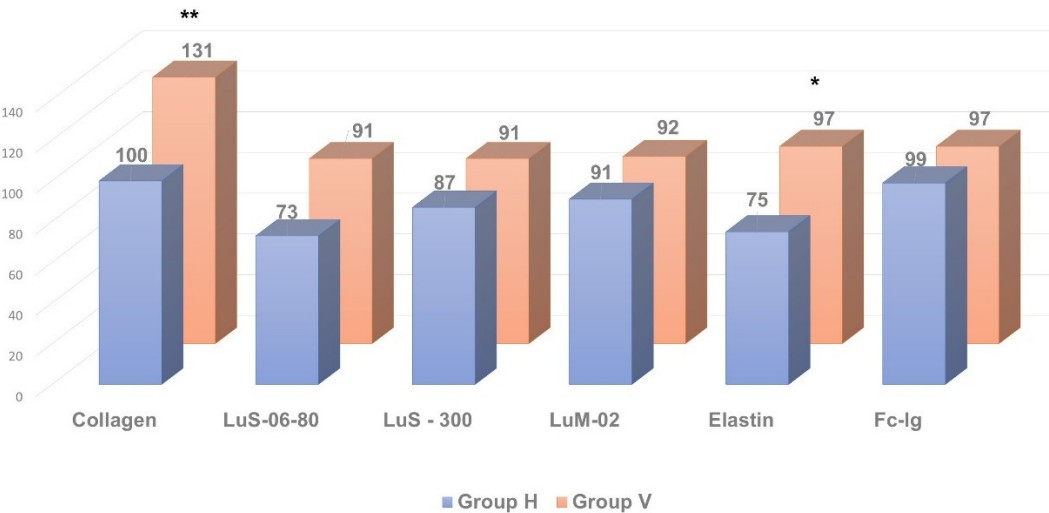

**Figure 3.** Comparative analysis of anti-pulmonary autoantibody concentration in individuals with clinically significant chronic fatigue of post-viral etiology (group V, n = 13) and healthy individuals (group H, n = 12). * $p \le 0.05$, ** $p \le 0.01$, group V > group H.

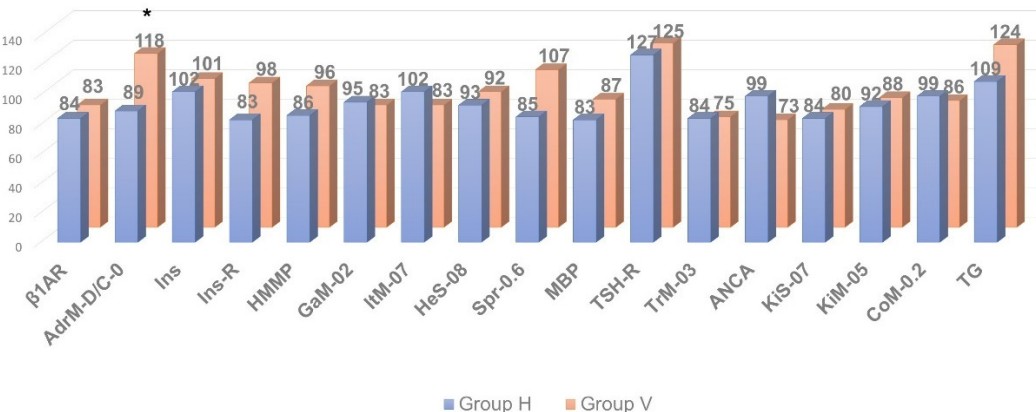

**Figure 4.** Comparative analysis of anti-visceral autoantibody concentration in individuals with clinically significant chronic fatigue of post-viral etiology (group V, n = 13) and healthy individuals (group H, n = 12). * $p \leq 0.05$, group V > group H.

Thus, in post-viral CFS/ME patients (group V) the levels of AAb towards several neural tissue antigens were significantly higher compared to the healthy controls (group H), though levels of AAb towards organ-specific visceral antigens were generally comparable to those in the control group, with the exception of the AAb towards apoptotic/debris products, that appeared to be higher in the post-viral group than in healthy controls. It is likely that an excessive autoimmune reaction to the neural/neuroendocrine targets and to the non-organ-specific antigens plays an important role in the pathogenesis of CFS/ME, although many complaints of the patients suffering from CFS/ME are related to the various seemingly somatic dysfunctions. Among all organ-specific AAb, group V had elevated levels of AAb in the adrenal medulla (which is also a paraganglion of the autonomic nervous system). It may allude that, in CFS/ME, visceral dysfunctions are secondary and result from primary altering autonomic nervous and neuroendocrine regulation. Hence, CSF/ME has the features that may relate it to autoimmune dysautonomia.

Figure 5 shows that the profile of AAb to the autoantigens of neural tissue and to non-organ-specific autoantigens (beta-2-glycoprotein I and double-stranded DNA) differs from normal levels in all presented types of clinically significant chronic fatigue, but the vector and degree of changes are not the same in various etiological subgroups. The most important differences are as follows:

- AAb to beta-2 glycoprotein-1 were increased in all cases of clinically significant chronic fatigue, but not in those individuals, who had complaints of non-CFS recurrent fatigue ("healthy but tired" group, HTd);
- Only post-viral asthenia (group V) is distinguished by a statistically significant increase in the level of AAb to voltage-dependent calcium channels, while the rise in the level of AAb to a number of autoantigens is the highest in post-viral chronic fatigue in comparison with other types of fatigue ($p < 0.05$);
- Only stress-related asthenia (group D) is characterized by a statistically significant increase in the level of autoantibodies to glutamate receptors;
- All types of fatigue, including acute recurrent subclinical fatigue, inappropriate to CFS/ME (group HTd), are characterized by an increase in the level of autoimmunity to the serotonin receptors and proteins GFAP and S-100, without significant differences between fatigue groups;
- No type of fatigue (neither positive, nor negative according to the CFS/ME criteria) is associated with an increase in autoimmune reactivity against the myelin basic protein.

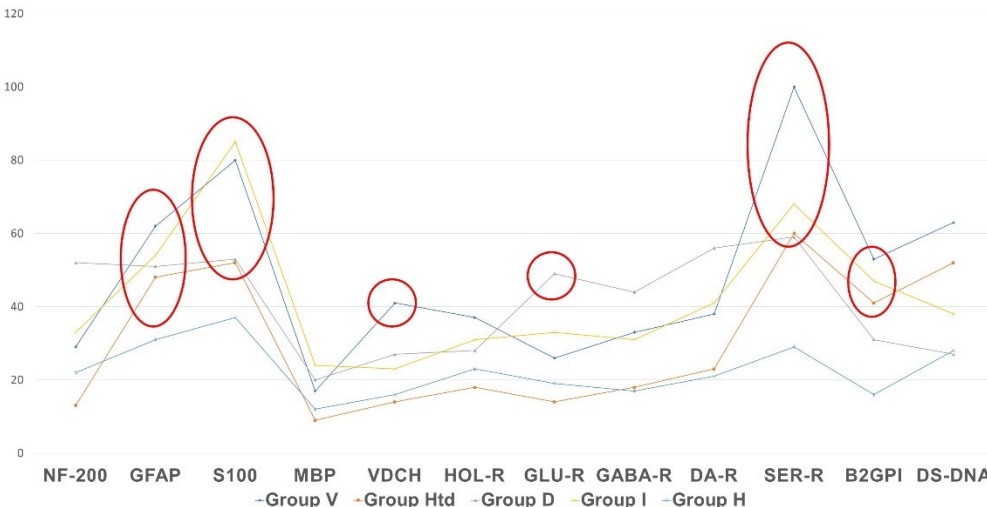

**Figure 5.** Profile of autoimmune reactivity to the neural and non-organ-specific visceral autoantigens in clinically significant chronic fatigue of various etiology, in comparison with control group. The absolute level of autoimmune reactivity in units of optical density is shown by the Y-axis. Statistically significant intergroup differences are outlined with ellipses or circles.

The study of the correlation between an increase in the absolute level of various AAb and the presence of the confirmed CFS/ME diagnosis was performed by comparing the autoimmune profile of 24 subjects who meet the CFS/ME criteria with a group of clinically healthy donors (group H, table 2).

**Table 2.** The risk of developing or not developing chronic fatigue syndrome with a certain autoantibody evaluation. VS, very strong correlation; S, strong correlation; M, moderate strength correlation; W, weak correlation; "+", positive correlation; and "-", negative correlation.

| Antibody | Correlation Values | *p*-Value |
|---|---|---|
| β2GPI | +0.792 (VS) | <0.001 |
| GFAP | +0.492 (S) | 0.006 |
| VDCh | +0.458 (S) | 0.011 |
| Hol-R | +0.385 (M) | 0.036 |
| Ser-R | +0.305 (M) | 0.119 |
| GABA-R | 0.187 (W) | 0.654 |
| MBP | Irrelated | 1 |
| Da-R | - 0.201 (W) | 0.500 |
| S100 | - 0.328 (M) | 0.081 |
| Glu-R | - 0.377 (S) | 0.044 |
| NF-200 | - 0.612 (VS) | <0.001 |

The most significant correlation was found between the risk of CFS/ME and the elevation of AAb to β-2 glycoprotein-1. The AAb towards GFAP, N-cholinergic receptors and voltage-dependent calcium channels, as well as AAb against serotonin receptors, correlated less strongly with the diagnosis of CFS/ME. AAb to γ-aminobutyric acid or to dopamine receptors weakly correlated with the diagnosis of CFS/ME. AAb to MBP were not associated with the diagnosis of CFS/ME at all. Of note, some AAb that were discordant with the diagnosis of CFS were also found. For example, the levels of AAb to the glutamate receptor and especially to the neurofilament protein NF200 were negatively associated with the presence of CFS/ME, thus characterizing AAb of this specificity as anti-risk factors for CFS/ME.

## 4. Discussion

Nowadays, the role of autoimmunity in the pathogenesis of CFS/ME is tackled in many studies [6–8,12]. The identification of autoantibodies in nerve tissue and autonomic receptors in such patients may play an important role in the development of effective diagnostic and therapeutic protocols. It is especially interesting that, in our study, the elevated levels of autoantibodies to β2-glycoprotein-I correlated most significantly with CFS/ME diagnosis. This particular autoantigen is also known as an antiphospholipid syndrome (APS) marker [37]. As early as 1999, a hypothesis was introduced considering CFS/ME as a form of mild chronic APS, confirmed by some similarities of hemostasis/antihemostasis system status in these two entities [38]. Our data can probably draw new attention to this old concept, being for a long time out of the scope of attention in CFS/ME studies.

There is also significant correlation of CFS/ME with AAb against some neural antigens, but in contrast, a significant negative correlation of this diagnosis with some other anti-neural antibodies (e.g., glutamate receptors and protein NF200). These data may be essential for the differential diagnosis of CFS/ME and other types of fatigue-associated diseases, including autoimmune ones.

Autoimmunity is inherent both in health and disease, in the first case it occurs as low regulatory titers of AAb, in the second, it manifests with pathologically increased concentrations of Aab reaching pathogenicity [26,35,36]. Our data suggests that the level of some AAb is elevated in patients with recurrent fatigue, but this is not specific for CFS/ME. AAb to the serotonin receptors and to the glial fibrillar acidic protein were evaluated in the study. Interestingly, earlier the impaired expression of the serotonin transporter and signs of neuroglia activation were demonstrated in the experimental model of CFS/ME obtained by rat immunization with polyribonucleotides [18].

In our study the links were established between certain AAb and some features of the etiology of chronic fatigue. Many types of AAb to antigens expressed in the nervous tissue, but only anti-adrenal medullar AAb, among all checked types of anti-visceral organ-specific AAb, significantly correlated with chronic fatigue. This fact suggests that the numerous complaints of such patients related to visceral dysfunctions are associated not with a direct autoimmune lesion of the internal organs, but with their secondary involvement mediated through autoimmune neuroendocrine dysregulation and/or dysautonomia [39].

In post-viral and stress-associated forms of clinically significant chronic fatigue autoimmune reactions against non-organ-specific antigens associated with apoptotic processes and tissue debris were evaluated. It may indicate the role of impaired clearance of apoptotic material and tissue debris in the pathogenesis of symptoms associated with CFS/ME, similar to the occurrence of this phenomenon of apoptotic clearance deficiency in lupus and other rheumatological diseases [40–43].

## 5. Conclusions

The autoimmunity profile studies in CFS/ME becomes more and more important, because millions of people already do suffer and even more will be suffering from post-COVID syndrome, which shares many common features with CSF/ME [44] and is considered as a result of enhanced autoimmune processes triggered by novel coronavirus infection [45]. Increased autoimmune reactions to the multiple neural antigens and to adrenal medullar antigen, but not to other tissue-specific somatic ones, were revealed in this study. An increase in autoantibody levels towards some neural and non-tissue specific antigens strongly correlated with a CFS/ME diagnosis. Autoimmune reactions were described in all subtypes of the clinically significant chronic fatigue. Visceral complaints in CFS/ME patients may be secondary to the neuroendocrine involvement and autoimmune dysautonomia. CFS may be closely interrelated with antiphospholipid syndrome, that requires further study.

## 6. Limitations

There are potential limitations/caveats to the current study. The control serum used is a preparation of polyclonal immunoglobulins of the IgG class, synthesized by B-lymphocytes in response to antigenic stimuli that occurred throughout the life of donors. Control serum immunoglobulins were obtained from the pooled blood serum of more than 5000 healthy donors and brought to a concentration close to physiological (16 mg/mL). Consequently, this sample contains population-normalized IgG class polyclonal antibodies, of which, many are relevant to the studied antigens. This allows us to use this control sample as a type of universal standard for the antigens in the test. Depending on the studied antigen, the control sample is diluted to a final concentration, which is calculated (derived) on the basis of studies of the level of autoantibodies of a large cohort of healthy people (individual serum samples). The reaction of the control sample in individual dilutions with different antigens reflects the individual profile of a healthy person in the population in the corresponding age group. When comparing the values of the parameters of the test sample (patient) with the control sample, we obtain a profile of deviations in the content of an individual's autoantibodies from the population 'norm'. Running individual controls rather than pooled controls represents an alternative validated approach. However, in the current study we have relied on pooled samples and recognize there are many possible control strategies that could be performed.

**Author Contributions:** Conceptualization, methodology, review and editing: L.P.C. ; formal analysis, writing: O.V.D., N.Y.G. All authors have read and agreed to the published version of the manuscript.

**Funding:** This work is supported by the grant of the Government of the Russian Federation for the state support of scientific research carried out under the supervision of leading scientists, agreement 14.W03.31.0009.: 14.W03.31.0009.

**Institutional Review Board Statement:** The study was approved by the Independent Ethical Committee of the St. Petersburg Scientific Research Institute of Phthisiopulmonology (extract from protocol No. 34.2 dated 01/19/2017) and the Local Ethical Committee of St. Petersburg State University (protocol No. 01-126 30.06.17). All study participants signed an informed consent. We confirm that all methods were performed in accordance with the relevant guidelines and regulations.

**Informed Consent Statement:** Informed consent was obtained from all subjects involved in the study.

**Data Availability Statement:** The datasets generated during and/or analysed during the current study are available from the corresponding author on reasonable request.

**Conflicts of Interest:** The authors declare no conflict of interest.

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
