# Peer review of "Chronic Fatigue Exhibits Heterogeneous Autoimmunity Characteristics Which Reflect Etiology"

_pathophysiology, doi:10.3390/pathophysiology29020016_

Round 1

Reviewer 1 Report

In this article entitled “Chronic Fatigue Shows Different Autoimmunity Spectra De-pending on Etiology” the authors evaluated autoimmune profiles in 33 patients with chronic fatigue of different origin compared to 20 healthy donors to unveil the pathogenesis of this disease. The studies are well controlled and rigorously conducted. The present article is engaging, and is of potential interest to the readers of Pathophysiology. I recommend the publication of this work only after the authors have addressed the following minor comment:

-In line 338 the authors should cite other research related to apoptotic clearance deficiency in rheumatological diseases (doi: 10.1186/s13075-019-1818-x), (doi.org/10.1038/nm939) and (doi: 10.1038/nrrheum.2015.8).

- The authors should consider adding a brief description in the introduction of APS and they should underline a putative link with CFS/ME.

- The authors should divided the long paragraph of Results in: Results; Discussion and Conclusions. It would be helpful to offer the reader in Conclusions a list of unresolved questions and future directions in CFS/ME research.

 -In addition, all figures should be provided in higher resolution.

Author Response

Dear Sir or Madame,

Please, find our deepest gratitude for your efforts and time, which makes our paper better!

1.  The recommended researches related to apoptotic clearance deficiency in rheumatological diseases were cited appropriately.

2. TThe brief description in the introduction of APS was provided in a separate paragraph and a putative link with CFS/ME was underlined.

3. The Results section was divided into Results; Discussion and Conclusions with some additional information added there.

4. All figures were provided in the doubly enlarged resolution.

Once again, thank you for your efforts,

Best regards, authors.

Reviewer 2 Report

It is inappropriate to use pooled sera from the patients vs controls to do the Ab studies. It would be appropriate, and potentially  interesting to compare the frequency of these auto-antibodies in the individual sera of a much larger group of individuals with CFS vs  age matched clearly normal individuals, say 75 patients vs 50 normals

Author Response

Dear Sir or Madam,

Please, find our deepest gratitude for your efforts and time, which makes our paper better!

Considering your valuable remark, we've consulted with prof. Leonid Churilov, who is also a co-author of this paper and a scientific advisor of dr. Danilenko. He asked to send the following response to your kind attention:

"The use of a reference pool of sera is a widespread practice in screening studies. In the attachments for this letter, you can kindly find some of the published works with a similar design. 
Note that conclusions are based not only on comparison with the pool of sera but also on absolute antibody concentrations, calculated in absorbance units. And that there are internal control groups - absolutely healthy without complaints and without diagnoses, but complaining of periodical fatigue. And they were examined adequately - by the same methods. and give statistically significant differences".

In addition, all data related to NEURO-panel (and this is 2/3 of the research) represented only in the absolute concentration of antibodies, calculated in units of optical density.

Also, we added in the study 30 cases of different etiology of CFS and 20 people of the control group (absolutely healthy and healthy but a little tired). Therefore, the ratio is the following: 50/75 = 66%; 20/30 = 66%. It's just that we have fewer people, since pathology in its pure form, with careful exclusion of all possible other causes of fatigue, is rare.

Once again, thank you very much for your conserns and efforts,

Best regards, authors.

Round 2

Reviewer 2 Report

Authors have not addressed my concerns about the methodology they used , especially pooling, which is not standard methodology when looking at the role of specific antibodies in autoimmune conditions etc--in addition, rather study autoantibodies known to be associated with specific autoimmune diseases.

Author Response

Dear Sir or Madame,

We are deeply grateful for your kind response and comments.

As was mentioned in the previous round, we fully understand your concerns on methodology of the paper. However, there are plenty of papers, where this method was applied. 

https://www.academia.edu/33518325/6_Self_recognition_self_interaction_and_physiologic_autoimmunity_Immune_system_and_microbes

Immunophysiology versus immunopathology: Natural autoimmunity in human health and disease - ScienceDirect

https://pubmed.ncbi.nlm.nih.gov/12965177/

https://pubmed.ncbi.nlm.nih.gov/18191542/

https://www.researchgate.net/publication/275634451_The_Natural_Autoimmunity_Self-Recognition_Self-Interaction_and_Self-Maintenance

https://www.researchgate.net/publication/275634451_The_Natural_Autoimmunity_Self-Recognition_Self-Interaction_and_Self-Maintenance

http://www.sryahwapublications.com/archives-of-immunology-and-allergy/pdf/v2-i2/7.pdf

https://www.em-consulte.com/article/836456/the-reactivity-of-human-serum-natural-autoantibodi

In addition, the absolute, calculated by ELISA and compared with each other (and not with a pool) autoantibody levels were mentioned in the paper, in particular in that part, which comes after the comparison of different fatigue groups and describes the ELI-neuro-test.

You also may kindly address prof. Leonid Churilov to have his comments on the methodology. He asked me to mention the following: “we are extremely grateful for presenting your views on methodology, but in the field of studying the profile of autoimmune reactivity (not searching for diagnostic markers of a particular disease, but specifically the profile of autoimmune reactivity in people who are not yet ill) - different authors may have different methodological approaches”.

Best regards, Natalia